# Moderation by better sleep of the association among childhood maltreatment, neuroticism, and depressive symptoms in the adult volunteers: A moderated mediation model

Jiro Masuya[1]*, Chihiro Morishita[1], Miki Ono[1], Mina Honyashiki[1], Yu Tamada[1], Tomoteru Seki[1], Akiyoshi Shimura[1], Hajime Tanabe[2], Takeshi Inoue[1]

1 Department of Psychiatry, Tokyo Medical University, Tokyo, Japan, 2 Faculty of Humanities and Social Sciences, Shizuoka University, Shizuoka, Japan

* j-masuya@tokyo-med.ac.jp

## Abstract

### Background

Previously, we demonstrated that childhood maltreatment could worsen depressive symptoms through neuroticism. On the one hand, some studies report that sleep disturbances are related to childhood maltreatment and neuroticism and worsens depressive symptoms. But, to our knowledge, no reports to date have shown the interrelatedness between childhood maltreatment, neuroticism, and depressive symptoms, and sleep disturbance in the one model. We hypothesized that sleep disturbance enhances the influence of maltreatment victimization in childhood or neuroticism on adulthood depressive symptoms and the mediation influence of neuroticism between maltreatment victimization in childhood and adulthood depressive symptoms.

### Subjects and methods

Total 584 Japanese volunteer adults recruited through convenience sampling from 4/2017 to 4/2018 were assessed regarding their characteristics of demographics, history of childhood maltreatment, sleep disturbance, neuroticism, and depressive symptoms with questionnaires self-administered. Survey data were analyzed using simple moderation models and a moderating mediation model.

### Results

The interaction of sleep disturbance with childhood maltreatment or neuroticism on depressive symptoms was significantly positive. Furthermore, the moderating effect of sleep disturbance on the indirect effect of childhood maltreatment to depressive symptoms through neuroticism was significantly positive.

### Limitations

Because this was a cross-sectional study, a causal relationship could not be confirmed.

**Data Availability Statement:** Data cannot be shared publicly because of Ethics Committee

restriction. All relevant data are within the paper. Data are available from the Internal Review Board of the Department of Psychiatry, Tokyo Medical University (Japan) (contact via email: seisinka@tokyo-med.ac.jp) for researchers who meet the criteria for access to confidential data.

**Funding:** This work was partly supported by a Grant-in-Aid for Scientific Research (no. 21K07510, to TI) from the Japanese Ministry of Education, Culture, Sports, Science and Technology (https://www.jsps.go.jp/english/egrants/). We have corrected the information in the revised manuscript. The funders had no role in study design, data collection and analysis, decision to publish, or preparation of the manuscript.

**Competing interests:** I have read the journal's policy and the authors of this manuscript have the following competing interests: Jiro Masuya has received personal compensation from Otsuka Pharmaceutical, Eli Lilly, Astellas, MSD, Janssen Pharmaceutical, Takeda Pharmaceutical, Shionogi Pharmaceutical, and Meiji Yasuda Mental Health Foundation, as well as grants from Pfizer, and Mitsubishi Tanabe Pharma. Yu Tamada received honoraria from Otsuka Pharmaceutical, Sumitomo Pharma, Eisai, MSD and Meiji Seika Pharma. Takeshi Inoue has received personal fees from Mochida Pharmaceutical, Takeda Pharmaceutical, Eli Lilly, Janssen Pharmaceutical, MSD, Taisho Toyama Pharmaceutical, Yoshitomiyakuhin, and Ono Pharmaceutical; a grant from Astellas; and grants and personal fees from Otsuka Pharmaceutical, Sumitomo Dainippon Pharma, Mitsubishi Tanabe Pharma, Kyowa Pharmaceutical Industry, Pfizer, Shionogi, Tsumura, Novartis Pharma, Eisai, Daiichi Sankyo, and Meiji Seika Pharma; and is a member of the advisory boards of Pfizer, Novartis Pharma, and Mitsubishi Tanabe Pharma. All other authors declare that they have no actual or potential conflicts of interest associated with this study. This does not alter our adherence to PLOS ONE policies on sharing data and materials.

## Conclusions

Our findings indicate that individuals with milder sleep disturbance experience fewer depressive symptoms attributable to neuroticism and childhood maltreatment. Additionally, people with less sleep disturbance have fewer depressive symptoms arising from neuroticism owing to childhood maltreatment. Therefore, improvement of sleep disturbance will buffer the aggravating effect of childhood maltreatment, neuroticism caused by various factors, and neuroticism resulting from childhood maltreatment on depressive symptoms.

## Introduction

It is widely accepted that childhood maltreatment (CM) is one of the risk factors for the onset or worsening of depressive symptoms in adults [1, 2]. Because of the time gap between the occurrence of CM and the development of adulthood depressive symptoms, there are assumed to be some mediators. Increasing lines of evidence demonstrate that psychological characteristics such as cognitive styles or personality characteristics, mediate the relationship between CM and adulthood depression [3–5]. We previously reported that CM might aggravate depressive symptoms through neuroticism in the general population [6]. A narrative review defined neuroticism as "the tendency to experience frequent, intense negative emotions associated with a sense of uncontrollability (the perception of inadequate coping) in response to stress", with an underlying mechanism involving dynamic genetic and environmental interactions [7]. In addition, neuroticism is a risk factor for psychiatric problems, and has been specified as an essential risk factor for the development of major depressive disorder in the Diagnostic and Statistical Manual of Mental Disorders, Fifth Edition, revised in 2013 [8, 9].

Therefore, it is an important task to identify the negative influences of CM and neuroticism on adulthood depressive symptoms. Although few intervention studies have investigated treatments focusing on the aggravating influences of CM on adulthood depressive symptoms, Joss et al. reported that there was no group by time interaction between subjects with mindfulness-based intervention and a waiting list control group for changes of depressive symptoms in subjects who have experienced mild-to-moderate CM [10]. In contrast, many studies have reported that transdiagnostic cognitive behavioral therapy and paroxetine significantly improve neuroticism [11, 12]. Regarding observational studies, a recent prospective cohort study demonstrated that CM causes sleep disturbance, which results in the worsening of depressive symptoms, suggesting that the improvement of sleep disturbance may ameliorate the exacerbating effect of CM on adulthood depressive symptoms [13]. Additionally, we demonstrated that some personality traits interact with the influence of sleep disturbance on adulthood depressive symptoms. Our previous study showed that resilience, a personality trait that enables growth in the face of adversity, is the opposite of neuroticism and significantly alleviates sleep disturbances and depressive symptoms in adulthood [14, 15]. Furthermore, we showed that affective temperaments, a personality trait defined by Kraepelin as the fundamental state of manic-depressive illness and modified by Akiskal, significantly interact with the influence of sleep disturbance on adulthood depressive symptoms [16, 17].

But, to our knowledge, no study to date has researched the moderation effect of sleep for CM or neuroticism on adulthood depressive symptoms. Investigating this missing link would contribute to the treatment for depressive symptoms caused by CM and neuroticism. Therefore, we hypothesized the following: (1) sleep disturbance exacerbates the influence of CM on

depressive symptoms, (2) sleep disturbance worsens the influence of neuroticism on depressive symptoms, and (3) better sleep reduces the mediation influence of neuroticism between CM and depressive symptoms. We tested these hypotheses by simple moderation analyses and a moderated mediation analysis.

## Subjects and methods

### Subjects

Total 1,237 Japanese non-clinical volunteer adults recruited through convenience sampling at Tokyo Medical University from 4/2017 to 4/2018. Self-completion questionnaires were distributed to all subjects. Participants had to be at least 20 years old and give written consent to participate. Exclusion conditions were the presence of serious physical or organic psychiatric illness. Furthermore, all subjects were informed that participation was voluntary, that there would be no disadvantages for non-participation, and that anonymity would be ensured. Total 597 participants provided informed consent and replied to the survey. We excluded 13 individuals with missing data and analyzed data from a 584 total. The subjects' mean age was 41.7 ± 12.1 years, and 249 (42.6%) were male. This study was conducted in accordance with the Declaration of Helsinki (as amended in Fortaleza during 2013). It was also approved by the Institutional Review Board of Tokyo Medical University (research approval number: SH3502).

### Questionnaires

**Eysenck Personality Questionnaire-Revised (EPQ-R).** Neuroticism was assessed by the neuroticism subscale (12 items) of the shortened version of the EPQ-R, following the method of Kendler et al. [18, 19]. The reliability and validity of the Japanese short-scale version of the EPQ-R have been confirmed [20]. Each item is rated on a two-point scale. Higher total scores are interpreted as higher neuroticism.

**Patient Health Questionnaire-9 (PHQ-9).** PHQ-9 is a self-completion questionnaire for the measuring the severity of depressive symptoms [21]. The Japanese version was developed, and its high reliability and validity have been verified [22]. PHQ-9 is composed of nine questions, each of which is assessed using a four-point Likert scale, with higher total scores indicating more severe depressive symptoms.

**Child Abuse and Trauma Scale (CATS).** CATS is a self-completion questionnaire for evaluating abuse and related trauma in childhood [23]. The Japanese version was developed, and its high reliability and validity have been verified [24]. There are 3 subscales consisting of 38 items. The items were rated on a 5-point Likert scale from 0 to 4, and the total score was used for the analysis. Higher total scores are interpreted as indicating that the victimization of childhood abuse was more severe.

**Pittsburgh Sleep Quality Index Japanese Version (PSQI-J).** Pittsburgh Sleep Quality Index (PSQI) is a self-administered questionnaire for assessing sleep disturbance, and its high reliability and validity have been verified [25]. PSQI is composed of seven subscales. Each subscale is scored on a scale of 0 to 3. A higher total score is interpreted as poorer sleep quality and more severer sleep disturbance [25]. PSQI was translated to the Japanese version (PSQI-J), and validated by back translation [26].

### Data analysis

Pearson's correlation analyses were performed to investigate associations among age, CATS total scores, EPQ-R scores, PHQ-9 scores, and PSQI-J total scores. Our previous study using the different sample from this study showed that (1) CATS scores significantly directly and

indirectly positively predict increasing PHQ-9 scores via EPQ-R scores, and (2) EPQ-R scores significantly positively predict increasing PHQ-9 scores [6]. Therefore, this simple mediation analysis was not performed in the present study. Simple moderation analyses were performed to investigate the hypothesis that PSQI-J total scores moderate the effect of CATS total scores and EPQ-R scores on PHQ-9 scores. Finally, moderated mediation analysis was performed according to the hypothesis that PSQI-J total scores moderate the mediation effect of EPQ-R scores from CATS total scores to PHQ-9 total scores.

All analyses were performed with R version 4.1.2. software (R core team, Vienna, Austria), and PROCESS for R macro version 4.1 software was used for the simple moderation analyses (Model 1 in PROCESS) and the moderated mediation analysis (Model 15 in PROCESS) (Hayes., 2022). Variables used as the interaction terms were centralized before the analysis. 95% bias-corrected confidence intervals of the indirect effect, and the index of moderated mediation in the moderated mediation model were computed using bootstrapping methods (n = 2,000). An indirect effect was considered significant if the 95% confidence interval (95% CI) for the indirect effect did not contain 0. In addition, when the index of moderated mediation did not include 0, the moderated mediation was judged to be significant. Unstandardized partial coefficients rather than standardized partial coefficients were reported in this study, because PROCESS version 4.1 was not able to compute precise standardized partial coefficients of variables that are calculated using bootstrapping methods in moderated mediation analyses. All the analyses included age and sex as covariates. The statistical significance level was defined as a $p$-value of less than 0.05.

## Results

### Demographic information (Table 1)

Data are presented as means ± standard deviations (SD) or numbers.
Table 1 presents the demographic information of the 584 subjects.

### Pearson's correlation analysis (Table 2)

The associations among age, CATS total scores, EPQ-R scores, PHQ-9 scores, and PSQI-J total scores are shown in Table 2. CATS total scores significantly and positively correlated with PHQ-9 scores, and PSQI-J total scores. EPQ-R scores significantly and positively correlated with CATS total scores, PHQ-9 scores, and PSQI-J total scores, whereas they significantly and

**Table 1. Demographic characteristics of the subjects.**

| Characteristic or measure | Number or mean ± SD |
|---|---|
| Age (years) | 41.7 ± 12.1 |
| Sex (men: women) | 249 (42.6%): 335 (57.4%) |
| Years of education | 14.6 ± 1.8 |
| Employment status (employed: nonemployed) | 555: 25 |
| Current marital status (married: single) | 382: 197 |
| Presence of offspring (yes: no) | 365: 215 |
| Living alone (yes: no) | 112: 459 |
| Past history of mental illness (yes: no) | 68 (11.6%): 516 (88.4%) |
| Current mental illness (yes: no) | 26: 548 |

**Table 2. Pearson's correlation coefficients (*r*) among age, CATS total scores, EPQ-R scores, PHQ-9 scores, and PSQI-J total scores of the subjects.**

|  | Mean | SD | Age | CATS | EPQ-R | PHQ-9 | PSQI-J |
|---|---|---|---|---|---|---|---|
| Age | 41.7 | 12.1 | – |  |  |  |  |
| CATS | 27.3 | 20.0 | 0.0727 | – |  |  |  |
| EPQ-R | 4.4 | 3.5 | −0.196*** | 0.225*** | – |  |  |
| PHQ-9 | 4.1 | 4.2 | 0.0344 | 0.320*** | 0.558*** | – |  |
| PSQI-J | 5.8 | 3.5 | 0.0315 | 0.213*** | 0.321*** | 0.566*** | – |

***$p < 0.001$

CATS, Child Abuse and Trauma Scale; EPQ-R, Eysenck Personality Questionnaire-Revised; PHQ-9, Patient Health Questionnaire-9; PSQI-J, Pittsburgh Sleep Quality Index Japanese Version; SD, standard deviation

negatively correlated with age. Additionally, PHQ-9 scores significantly and positively correlated with PSQI-J total scores.

## Simple moderation analyses

**Interaction of CATS total scores and PSQI-J total scores on PHQ-9 scores.** CATS total scores, PSQI-J total scores, and the interaction term obtained by multiplying these 2 variables significantly and positively correlated with PHQ-9 scores (F [2,546] = 81.43, p < 0.001; CATS total score, unstandardized partial coefficient B = 0.044, p < 0001; PSQI-J global score, B = 0.626, p < 0.001; interaction term, B = 0.005, p = 0.015). The adjusted $R^2$ was 0.376, indicating that 37.6% of the variance in PHQ-9 scores was accounted for by the model. The ΔR2 value, which indicates the difference of the $R^2$ of the models with and without the interaction term, was significantly positive.

**Interaction of EPQ-R scores and PSQI-J global scores on PHQ-9 scores.** EPQ-R scores, PSQI-J global scores, and the interaction term obtained by multiplying the former 2 variables significantly and positively associated with PHQ-9 scores (F [2,548] = 126.26, p < 0.001; EPQ-R score, B = 0.478, p < 0001; PSQI-J global score, B = 0.503, p < 0.001; interaction term, B = 0.042, p < 0.001). The adjusted $R^2$ was 0.481 and ΔR2 was significantly positive.

## Moderated mediation analysis (Table 3 and Fig 1)

Resuits of the moderated mediation analysis are presented in Fig 1 and Table 3. CATS scores were significantly positively associated with EPQ-R scores, and EPQ-R scores were significantly positively associated with PHQ-9 scores. The direct effect of CATS total scores on PHQ-9 scores was also significantly positive. Moreover, the interaction of PSQI-J global scores with CATS total scores and EPQ-R scores on PHQ-9 scores was significantly positive (Fig 1).

**Table 3. Conditional indirect effects of CATS total scores on PHQ-9 scores through EPQ-R across levels of PSQI-J global scores.**

| Moderator (PSQI-J) levels | | Effect | SE | 95% CI (lower, upper) |
|---|---|---|---|---|
| Low | (M− 1SD) | 0.013 | 0.004 | (0.007, 0.021) |
| Medium | (M) | 0.018 | 0.004 | (0.011, 0.027) |
| High | (M + 1SD) | 0.024 | 0.006 | (0.013, 0.035) |

CATS, Child Abuse and Trauma Scale; EPQ-R, Eysenck Personality Questionnaire-Revised; PHQ-9, Patient Health Questionnaire-9; PSQI-J, Pittsburgh Sleep Quality Index Japanese Version; M, mean; SD, standard deviation, SE, standard error, CI, confidential interval.

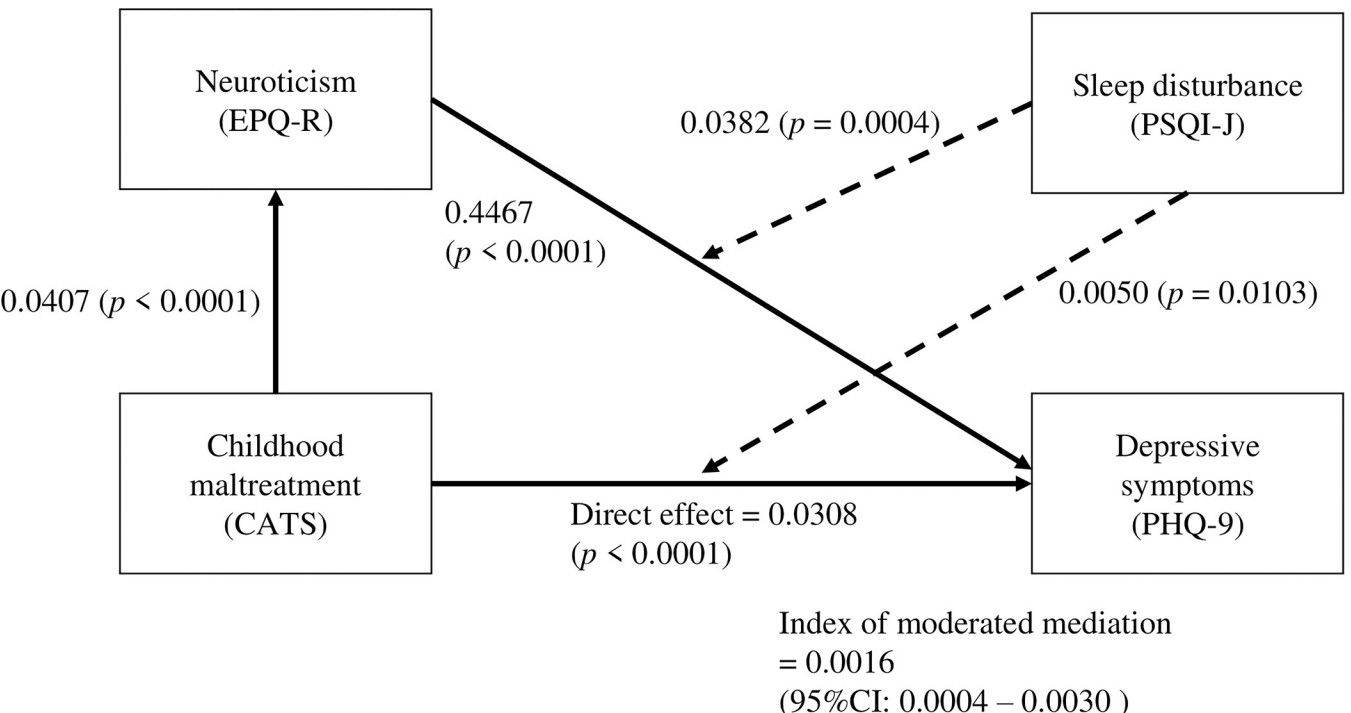

**Fig 1. Moderated mediation model of sleep disturbance, childhood maltreatment, neuroticism and depressive symptoms.** The dashed arrows indicate moderation effects. The numbers next to the arrows suggest unstandardized partial coefficients. CATS, Child Abuse and Trauma Scale; EPQ-R, Eysenck Personality Questionnaire-Revised; PHQ-9, Patient Health Questionnaire-9; PSQI-J, Pittsburgh Sleep Quality Index Japanese Version; CI, confidence interval.

In addition, the conditional indirect effects of CATS total scores via EPQ-R scores on PHQ-9 scores at 3 PSQI-J global scores (mean –1 standard deviation, mean, mean +1 standard deviation) were significantly positive, and the conditional indirect effects became higher as the PSQI-J global scores became higher (Table 3). In addition, the index of moderated mediation was positive significantly (Fig 1). The model was significant and the adjusted $R^2$ was 0.514.

## Discussion

We presented for the first time to our knowledge the significant interaction effects of sleep disturbance with CM and neuroticism on depressive symptoms in simple moderation models in this study. The moderated mediation model showed the significant moderation of sleep on the direct effects of CM and neuroticism on depressive symptoms, and the significant moderated mediation effect of CM on depressive symptoms via neuroticism, depending on the level of sleep disturbance. The present findings support our hypotheses. In other words, better sleep is associated with a smaller negative effect of CM or neuroticism on depressive symptoms, and a smaller aggravating mediation effect of CM on depressive symptoms through neuroticism.

Considering the results of the significant interaction of CM and sleep disturbance on depressive symptoms, an improvement of sleep could ameliorate the negative influence of CM on depressive symptoms. Sleep disturbance as a modifiable moderation factor is important because CM is not a factor that can be intervened in clinical practice for grown-ups. Sleep can be improved by treating sleep disorders or maintain sleep hygiene [27, 28]. Moreover, we showed the significant exacerbation of sleep disturbance on the direct effect of CM on depressive symptoms, suggesting that improving sleep can buffer the influence of CM on depressive

symptoms by a pathway other than improving neuroticism. To our knowledge, there are no similar reports regarding the moderation effect of CM and sleep on adult mental health.

Regarding the results of the significant interaction of neuroticism and sleep disturbance on depressive symptoms, improvement of sleep disturbance could buffer the effect of neuroticism on depressive symptoms. Previously, Zhou et al. showed that chronotype and sleep quality significantly moderate the influence of resilience on depressive symptoms [29]. Moreover, we confirmed the significant interaction between resilience to depressive symptoms and sleep disturbances using the same subjects as in the present study [15]. As noted in the Introduction chapter, resilience appears to be an opposite personality trait to neuroticism [30], suggesting that the above findings in association with resilience support our present results. Additionally, our results suggest that improvement of neuroticism could ameliorate the worsening effect of sleep disturbance on depressive symptoms. As noted in the Introduction chapter, previous studies reported that transdiagnostic cognitive behavioral therapy and paroxetine are effective for neuroticism, suggesting that these interventions ameliorate the depressive symptoms caused by sleep disturbance [11, 12]. On the other hand, in a previous study, a 3-year prospective study reported that the presence of workplace harassment increased neuroticism, and the absence of workplace harassment reduced neuroticism. This indicates that the neuroticism are not immutable dispositions. This suggests that stressors such as workplace harassment may further exacerbate depressive symptoms caused by sleep disturbances [31].

Finally, in this study, we demonstrated that better sleep moderates the mediation effect of CM on depressive symptoms via neuroticism. Barlow et al. proposed the theory of the developmental origins of neuroticism, in which they assumed that synergistic psychological and biological vulnerabilities underlie neuroticism [7]. In other words, the etiologies of neuroticism are heterogeneous, suggesting that focusing on the neuroticism caused specifically by CM is methodologically necessary.

In this study, we showed the moderating role of sleep disturbance on the effect of CM and neuroticism on depressive symptoms, and the effect of CM on depressive symptoms through neuroticism. In a clinical setting, assessment of CM damage, neuroticism, and sleep disturbances, and interventions to ameliorate sleep disturbances, may be useful in treating depressive symptoms in subjects with the victimization of CM and neuroticism.

## Limitations

There are several limitations to the present study. First, the subjects in this study were recruited using convenience sampling, which is not strict probability sampling. This reduces the randomness of subject selection, which may reduce the accuracy of the results by including potential bias. Second, we could not conclude the causal associations among CM, neuroticism, depressive symptoms, and sleep disturbance, because the present study was a cross-sectional study. Long-term prospective longitudinal studies are needed to confirm the causal associations between sleep disturbance and depressive symptoms. Third, subjects in this study were not confirmed to have depression or other psychiatric disorders. In addition, we did not investigate the presence or absence of an actual diagnosis of depression in the patients, as only a self-administered survey was used in this study. Therefore, the results of this study cannot be used as being associated with depression. Sleep disturbances are a well-known symptom of depression and a factor indicating the severity of depression. However, in the present study, we did not consider sleep disturbances as only a symptom of depression, but also as a risk factor for enhancing the effects of CM on depressive symptoms in adulthood, and the effects of neuroticism on depressive symptoms in adulthood [32, 33]. Fourth, in this study, sleep

disturbances were assessed subjectively and future research should be based on objective data. Finally, recall bias may have affected the subjects' history of CM.

## Conclusion

To our best knowledge, this study is the first to study to date to report the moderation effect of better sleep on the effect of CM, neuroticism and the mediation effect of CM via neuroticism on depressive symptoms. These findings suggest the importance of intervening with sleep disturbance for the treatment for depressive symptoms in subjects with CM or neuroticism. More extensive prospective studies are necessary to clarify these points in the future.

## Acknowledgments

We thank Dr. Nobutada Takahashi of Fuji Psychosomatic Rehabilitation Institute Hospital, Dr. Hiroshi Matsuda of Kashiwazaki Kosei Hospital, Dr. Yasuhiko Takita (deceased) of Maruyamasou Hospital, and Dr. Yoshihide Takaesu of Izumi Hospital for their collection of subject data. We thank Dr. Helena Popiel of the Center for International Education and Research, Tokyo Medical University, for editorial review of the manuscript.

## Author Contributions

**Conceptualization:** Jiro Masuya, Tomoteru Seki, Akiyoshi Shimura, Hajime Tanabe, Takeshi Inoue.

**Data curation:** Jiro Masuya, Chihiro Morishita, Takeshi Inoue.

**Formal analysis:** Jiro Masuya, Takeshi Inoue.

**Funding acquisition:** Jiro Masuya.

**Investigation:** Jiro Masuya, Miki Ono, Mina Honyashiki, Takeshi Inoue.

**Methodology:** Jiro Masuya, Akiyoshi Shimura, Takeshi Inoue.

**Project administration:** Jiro Masuya, Chihiro Morishita, Takeshi Inoue.

**Resources:** Jiro Masuya.

**Software:** Jiro Masuya.

**Supervision:** Jiro Masuya, Chihiro Morishita, Akiyoshi Shimura, Takeshi Inoue.

**Validation:** Jiro Masuya, Chihiro Morishita, Miki Ono, Mina Honyashiki, Yu Tamada, Tomoteru Seki, Hajime Tanabe.

**Visualization:** Jiro Masuya, Chihiro Morishita, Miki Ono, Mina Honyashiki, Yu Tamada, Tomoteru Seki.

**Writing – original draft:** Jiro Masuya, Chihiro Morishita, Miki Ono, Mina Honyashiki, Yu Tamada, Tomoteru Seki, Akiyoshi Shimura.

**Writing – review & editing:** Jiro Masuya, Chihiro Morishita, Tomoteru Seki, Hajime Tanabe, Takeshi Inoue.

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
