## [Decision Letter · Decision Letter 0]

12 Feb 2024

PONE-D-23-24222

Moderation by better sleep of the association among childhood maltreatment, neuroticism, and depressive symptoms in the adult volunteers: a moderated mediation model

PLOS ONE

Dear Dr. Masuya,

Thank you for submitting your manuscript to PLOS ONE. After careful consideration, we feel that it has merit but does not fully meet PLOS ONE’s publication criteria as it currently stands. Therefore, we invite you to submit a revised version of the manuscript that addresses the points raised during the review process.

We look forward to receiving your revised manuscript.

Kind regards,

Anthony A. Olashore, MBCHB, PhD, FWACP

Academic Editor

PLOS ONE

Journal Requirements:

Reviewers' comments:

Reviewer's Responses to Questions

**Comments to the Author**

1. Is the manuscript technically sound, and do the data support the conclusions?

Reviewer #1: Yes

Reviewer #2: Yes

2. Has the statistical analysis been performed appropriately and rigorously? 

Reviewer #1: I Don't Know

Reviewer #2: Yes

3. Have the authors made all data underlying the findings in their manuscript fully available?

Reviewer #1: Yes

Reviewer #2: Yes

4. Is the manuscript presented in an intelligible fashion and written in standard English?

Reviewer #1: No

Reviewer #2: Yes

5. Review Comments to the Author

Reviewer #1: The study has groundbreaking potential on interventions designed for childhood trauma and depression.

1. Please go through the manuscript and revise grammar. some sentences are too long.

2. In my opinion, the authors should acknowledge limitations of using self-administered instruments in their limitations

3. As mentioned under limitations, sleep disturbance is a well researched and documented symptom of depression; there is however evidence suggesting it as a risk factor as well. The authors should justify why in this study they chose to consider it as a risk factor only and not account for it as a symptom signifying severity of depression.

Reviewer #2: This is a very good paper. The authors mentioned the mean age of participants; however, I suggest that authors include the sociodemographic characteristics of participants (add a table summarising this). This will help readers when they want to make decisions in evidence-based medicine. They can compare their clients or patients with the study population for similarity (external validity of the paper).

Maybe the authors could also add non-probability sampling as a limitation of the study.

6. PLOS authors have the option to publish the peer review history of their article (what does this mean?). If published, this will include your full peer review and any attached files.

Reviewer #1: No

Reviewer #2: **Yes: **stephane tshitenge

---

## [Author Response · Author response to Decision Letter 0]

28 Apr 2024

REVIEWER #1

1. Please go through the manuscript and revise grammar. Some sentences are too long.

Response:

We thank you for your comments. In accordance with your suggestion, we have revised some sentences and asked a native English speaker experienced in proofreading scientific documents to recheck the paper.

2. In my opinion, the authors should acknowledge limitations of using self-administered instruments in their limitations

3. As mentioned under limitations, sleep disturbance is a well researched and documented symptom of depression; there is however evidence suggesting it as a risk factor as well. The authors should justify why in this study they chose to consider it as a risk factor only and not account for it as a symptom signifying severity of depression. 

Response:

We thank you for your comments. In accordance with the comment, we have revised the limitations section of the revised manuscript, as follows. 

“Third, subjects in this study were not confirmed to have depression or other psychiatric disorders. In addition, we did not investigate the presence or absence of an actual diagnosis of depression in the patients, as only a self-administered survey was used in this study. Therefore, the results of this study cannot be used as being associated with depression.” (page 19, lines 277 to 281)

“Sleep disturbances are a well-known symptom of depression and a factor indicating the severity of depression. However, in the present study, we did not consider sleep disturbances as only a symptom of depression, but also as a risk factor for enhancing the effects of CM on depressive symptoms in adulthood, and the effects of neuroticism on depressive symptoms in adulthood [32,33].” (page 19, lines 281 to 285)

REVIEWER #2

1. The authors mentioned the mean age of participants; however, I suggest that authors include the sociodemographic characteristics of participants (add a table summarising this). This will help readers when they want to make decisions in evidence-based medicine. They can compare their clients or patients with the study population for similarity (external validity of the paper).

Response:

We thank you for your comments. In accordance with your suggestion, we added a table with the demographic information of the 584 subjects, as Table 1 of the revised manuscript. (page 12, lines 173 to 174)

2. Maybe the authors could also add non-probability sampling as a limitation of the study. 

Response:

We thank you for your comments. In accordance with the comments, we have revised the limitations section, as follows. 

“First, the subjects in this study were recruited using convenience sampling, which is not strict probability sampling. This reduces the randomness of subject selection, which may reduce the accuracy of the results by including potential bias.” (page 19, lines 271 to 274)

---

## [Decision Letter · Decision Letter 1]

23 May 2024

Moderation by better sleep of the association among childhood maltreatment, neuroticism, and depressive symptoms in the adult volunteers: a moderated mediation model

PONE-D-23-24222R1

Dear Dr. Masuya,

We’re pleased to inform you that your manuscript has been judged scientifically suitable for publication and will be formally accepted for publication once it meets all outstanding technical requirements.

Kind regards,

Anthony A. Olashore, MD, PhD.

Academic Editor

PLOS ONE

Additional Editor Comments (optional):

Reviewers' comments:

Reviewer's Responses to Questions

**Comments to the Author**

1. If the authors have adequately addressed your comments raised in a previous round of review and you feel that this manuscript is now acceptable for publication, you may indicate that here to bypass the “Comments to the Author” section, enter your conflict of interest statement in the “Confidential to Editor” section, and submit your "Accept" recommendation.

Reviewer #1: All comments have been addressed

2. Is the manuscript technically sound, and do the data support the conclusions?

Reviewer #1: Yes

3. Has the statistical analysis been performed appropriately and rigorously? 

Reviewer #1: Yes

4. Have the authors made all data underlying the findings in their manuscript fully available?

Reviewer #1: No

5. Is the manuscript presented in an intelligible fashion and written in standard English?

Reviewer #1: Yes

6. Review Comments to the Author

Reviewer #1: (No Response)

7. PLOS authors have the option to publish the peer review history of their article (what does this mean?). If published, this will include your full peer review and any attached files.

Reviewer #1: No

---

## [Editor Report · Acceptance letter]

28 May 2024

PONE-D-23-24222R1 

PLOS ONE

Dear Dr. Masuya, 

I'm pleased to inform you that your manuscript has been deemed suitable for publication in PLOS ONE. Congratulations! Your manuscript is now being handed over to our production team.

Kind regards, 

on behalf of

Dr. Anthony A. Olashore 

Academic Editor

PLOS ONE